# Effect of Bariatric Surgery on Metabolic Diseases and Underlying Mechanisms

**DOI:** 10.3390/biom11111582

**Published:** 2021-10-26

**Authors:** Yu Ji, Hangil Lee, Shawn Kaura, James Yip, Hao Sun, Longfei Guan, Wei Han, Yuchuan Ding

**Affiliations:** 1Department of General Surgery, Beijing Luhe Clinical Institute, Capital Medical University, Beijing 101149, China; doc.jiyu@foxmail.com; 2Department of Neurosurgery, Wayne State University School of Medicine, Detroit, MI 48201, USA; hangil.lee@med.wayne.edu (H.L.); ga6668@wayne.edu (S.K.); gp5940@wayne.edu (L.G.); yding@med.wayne.edu (Y.D.); 3John D. Dingell VA Medical Center, 4646 John R Street (11R), Detroit, MI 48201, USA; 4Department of General Surgery, Wayne State University School of Medicine, Detroit, MI 48201, USA; jyip@med.wayne.edu; 5Central Laboratory, Beijing Luhe Clinical Institute, Capital Medical University, Beijing 101149, China; m18810253032@163.com; 6China-America Institute of Neuroscience, Beijing Luhe Hospital, Capital Medical University, Beijing 101149, China

**Keywords:** obesity, metabolic disorders, complications, mechanisms, cardiovascular diseases

## Abstract

Obesity is a highly prevalent public health concern, attributed to multifactorial causes and limited in treatment options. Several comorbidities are closely associated with obesity such as the development of type 2 diabetes mellitus (T2DM), cardiovascular and cerebrovascular diseases, and nonalcoholic fatty liver disease (NAFLD). Bariatric surgery, which can be delivered in multiple forms, has been remarked as an effective treatment to decrease the prevalence of obesity and its associated comorbidities. The different types of bariatric surgery create a variety of new pathways for food to metabolize in the body and truncate the stomach’s caliber. As a result, only a small quantity of food is tolerated, and the body mass index noticeably decreases. This review describes the improvements of obesity and its comorbidities following bariatric surgery and their mechanism of improvement. Additionally, endocrine function improvements after bariatric surgery, which contributes to the patients’ health improvement, are described, including the role of glucagon-like peptide-1 (GLP-1), fibroblast growth factors 19 and 21 (FGF-19, FGF-21), and pancreatic peptide YY (PYY). Lastly, some of the complications of bariatric surgery, including osteoporosis, iron deficiency/anemia, and diarrhea, as well as their potential mechanisms, are described.

## 1. Introduction

Obesity has become a major global epidemic, plaguing societies for the past several decades. Obesity contributes exponentially to society’s disease burden by increasing the risk of diabetes, cardiovascular disease, and liver disease. In addition to these comorbidities, obesity also precipitates metabolic syndrome. Since the liver is the central integrator of metabolism and, therefore, plays an essential role in maintaining proper health, it is unsurprising that a majority of nonalcoholic fatty liver disease (NAFLD) cases are associated with obesity, abnormal lipid metabolism, and insulin resistance.

Bariatric surgery is a newly developed treatment for obesity and associated comorbidities. Improvements in weight loss, diabetes, and cardiovascular health are well established. However, the mechanisms through which this occurs are not clear. Although the benefit of bariatric surgery is evident, the means of its achievement remain to be clarified. Clarification of bariatric surgery’s mechanism of benefit may unlock other treatment strategies and modalities previously unknown.

The present review describes the improvements of obesity and its comorbidities following bariatric surgery, as well as their mechanism of improvement. A brief overview of different types of bariatric surgeries is surveyed to observe how the variations in technique result in different mechanisms of improvement of patient health. Additionally, endocrine function improvements after bariatric surgery, which contribute to the patient’s health improvement, are described, including the role of glucagon-like peptide-1 (GLP-1), fibroblast growth factors 19 and 21 (FGF-19, FGF-21), and pancreatic peptide YY (PYY). Lastly, some of the complications of bariatric surgery, including osteoporosis, iron deficiency/anemia, and diarrhea, as well as their potential mechanisms, are described.

## 2. NAFLD, Diabetes and Obesity

NAFLD is defined as the presence of steatosis in more than 5% of hepatocytes with little or no alcohol consumption, which includes benign nonalcoholic fatty liver disease (NAFLD) and the more severe nonalcoholic steatohepatitis (NASH) [1,2]. NAFLD is the most common cause of cryptogenic cirrhosis [3]. About 30–50% of NASH patients develop cirrhosis in 10 years [4]. In the United States, 31% of patients are diagnosed with NAFLD, and this value has been increasing in the past two decades [5]. The increasing prevalence is likely related to the rise in obesity. The correlation among NAFLD, diabetes, and obesity has been well established. The prevalence of NAFLD is much higher among obese populations with metabolic syndrome (MetS) and type 2 diabetes mellitus (T2DM), ranging from about 50% to 90% [6,7,8,9]. NAFLD currently affects around 27% of the Asian adult population [10]. Notably, Asians are more likely to have central fat deposition despite their lower average BMI [11,12,13,14]. Therefore, obesity may be more harmful to Asian populations compared to other demographics. Medical modalities for treatment of obesity—such as a balanced low-calorie diet, anorectic drugs, behavioral therapy, and exercise—have virtually no effect on most morbidly obese patients. In these situations, bariatric surgery is considered to be the most effective treatment option [15]. 

## 3. Benefit of Bariatric Surgery

Failure of medical therapy for severe obesity and success of surgery have produced a remarkable series of new techniques and procedures in the field of bariatric surgery. The history of weight loss surgery dates back to 1953 [16] and innovation has continued for 60 years thereafter. There are six procedures in bariatric surgery that are performed most frequently (Figure 1): jejunoileal bypass (JIB), Roux-en-Y gastric bypass (RYGB), vertical banded gastroplasty (VBG), biliopancreatic diversion (BPD) with or without duodenal switch (DS), adjustable gastric banding (AGB), and sleeve gastrectomy (SG). RYGB is one of the most commonly performed bariatric procedures and has been shown to have lasting effects on weight loss in adolescent and adult patients. In addition to reductions in weight gain, RYGB has profound effects on lowering blood glucose, reducing circulating triglycerides, and abating T2DM [17].

The different surgical methods act through a variety of mechanisms. Not only are these mechanisms directly associated with weight loss, but they may also be related to structural and endocrine changes. Each type of surgery and its specific structural change are briefly described in Table 1. More specifically, the table includes the year of the procedure’s creation, involvement of the gastric body, bypass, biliopancreatic diversion, bowel anastomosis, and weight loss rates. 

Bariatric surgery has been considered as the most effective available therapy for obese patients with diabetes [18]. As metabolic syndrome and diabetes are critically linked to NAFLD, it is plausible to conclude that bariatric surgery has the potential to provide improvements in liver-related metabolic disease. Bariatric surgery has proven to be efficient in reducing mortality, cardiometabolic risks, and liver disease [19]. However, the mechanisms underlying the acute improvements in metabolism including glucose and lipid homeostasis are not well elucidated [20], which merits further research into this topic. 

### Types of Bariatric Surgery

Figure 1 denotes the various bariatric surgery techniques that have been performed. It is shown that the gastrointestinal system can change structurally as a result of a procedure. Jejunoileal bypass surgery (JIB) constructs a bypass to most of the small intestine in an obese patient undergoing bowel reconstructive surgery using an end-to-end anastomosis between the jejunum and ileum. Separate drainage of the bypassed bowel is also established by means of a partial removal of the colon and full removal of the ileum. In Roux-en-Y gastric bypass (RYGB), the most widely used bariatric surgery procedure, a 15 to 30 cm long pouch stemming from the proximal stomach is connected to a loop of the jejunum, creating an anastomosis between the stomach and a proximal part of the jejunum. The rest of the stomach and proximal small bowel are re-anastomosed 80–120 cm distal to the stomach and jejunum anastomosis, allowing for an isolated point for flow of nutrients. The vertical banded gastroplasty (VBG) procedure involves using a single band and multiple staples to create a small stomach pouch within the stomach. A 1 cm hole is placed at the bottom of the pouch, where the ingested nutrients can flow into the remainder of the stomach and then into the remainder of the gastrointestinal tract. In the adjustable gastric band (AGB) procedure, an inflatable band is placed around the superior portion of the stomach to inevitably create a smaller stomach pouch, which will ultimately inhibit hunger levels and reduce food intake. Biliopancreatic diversion with duodenal switch (BPD/DS) is now often performed over the original procedure known as the biliopancreatic diversion (BPD) due to its lower frequency of complications. In this modern procedure, a portion of the stomach is resected, creating a sizably smaller stomach. A distal portion of the small intestine is then connected to the remaining pouch, bypassing the duodenum and jejunum. Sleeve gastrectomy (SG) is a surgical weight-loss procedure in which the stomach’s size is reduced by roughly 15%. To achieve this, a large portion of the greater curvature of the stomach is resected. The resection originates from the antrum and runs up to the cardia portions of the stomach.

## 4. Consequences of Bariatric Surgery

### 4.1. Eating Habits

Regardless of which bariatric surgery is performed, changes in eating habits are inevitable. According to the 2014 European guidelines [21], patients who have undergone bariatric surgery are asked to adhere to a plan of multiple small meals each day to ensure proper levels of vitamin B12, 25(OH) vitamin D3, parathyroid hormone, bone alkaline phosphatase, ferritin, Ca, pre-albumin, albumin, transferrin, creatinine, prothrombin time (PTT), etc. While this plan of eating multiple meals may seem slightly contradictory, it is necessary as the residual stomach volume limits the patient’s food intake. While patients must consume enough nutrients to maintain healthy diet standards, they are unable to overeat. Therefore, surgery inevitably changes the patient’s dietary habits. These changes in eating patterns can affect hormone signaling pathways such as insulin, glucagon, ghrelin, and many others, which link nutrient metabolism to the liver’s internal clock [22]. In addition to modifying the timing of meals, the content of meals is recommended to change as well. The Mediterranean diet includes olive oil, which is rich in monounsaturated fat (MUFA), nuts, fruits and legumes, vegetables, and fish, in conjunction with a low intake of red meat, processed meats, and sweets. A plethora of evidence suggests that the Mediterranean diet is beneficial for growing a healthy metabolic profile [23]. This may also provide an explanation for the reduction in NAFLD after bariatric surgery.

### 4.2. Weight Loss

The most intuitive change after bariatric surgery is weight loss, the benefits of which are immense. A recent study [24] assessed how different levels of weight loss impacted metabolic function and adipose tissue biology. When weight loss was greater than 16% of initial weight, plasma free fatty acid (FFA) and C-reactive protein (CRP) concentrations decreased while plasma adiponectin level increased significantly. The study further found a preferential loss of intraabdominal and intrahepatic fat when compared to initial overall body fat. A weight loss of 3–5% reduced steatosis, while a ≥5–7% drop in weight resolved NASH. Greater reductions in weight (i.e., ≥10%) may also improve hepatic fibrosis [25]. The impact of weight loss on histological improvement relies on the degree of weight reduction irrespective of the method used to reach it [26].

### 4.3. Effect on Diabetes

Bariatric surgery decreases rates of T2DM by targeting multiple aspects of the disease. It results in improved glucose control and initiated remission of diabetes in 95–100% of patients [27]. The underlying mechanisms were originally thought to occur by decreasing body weight. Thus, theoretically, those who have the greatest decrease in bodyweight were considered to have the greatest improvement in T2DM [28]. However, more studies found that modifications in metabolism and hormones played a greater role in fighting diabetes. In other words, bariatric surgery has the potential to treat diabetes by increasing the likelihood of insulin independency, improving incretin secretion, recovering islet function, and restoring peripheral insulin sensitivity to regulate glucose homeostasis [29,30,31,32,33,34,35,36]. Furthermore, bariatric surgery decreases circulating succinate levels [37] and curbs Krebs cycle completion to avoid excess glucose production, resulting in multiple metabolic improvements.

Although T2DM patients overtly outnumber T1DM patients, significant progress has also been made in understanding the impact of bariatric surgery on T1DM and its associated biomarkers. One study indicated that the associated complications of T1DM are potentially curbed by bariatric surgery, similar to the benefit seen in T2DM patients [38]. Evidence suggests significant drops in insulin, glycosylated hemoglobin (HbA1c), net body mass index, triglycerides, cholesterol, and blood pressure after bariatric surgery. Although evidence is still limited due to a paucity of studies and early stages of trials, the existing results show bariatric surgery to be promising in diminishing rates of morbidities and mortalities caused by T1DM.

### 4.4. Effect on Cardiovascular and Cerebrovascular Diseases

Obesity is a risk factor for cardiovascular and cerebrovascular diseases, while bariatric surgery is protective of the heart and vasculatures. Bariatric surgery reduces the incidence of sudden large vascular diseases (defined as the first onset of acute myocardial infarction, unstable angina, percutaneous coronary intervention, coronary artery bypass grafting, ischemic stroke, hemorrhagic stroke, carotid stenting, or carotid endarterectomy) [39,40,41,42]. Astonishingly, many patients with pre-existing cardiovascular disease can reduce or completely discontinue their cardiovascular medications after undergoing bariatric surgery [43].

After bariatric surgery, blood triglyceride and glucose levels are significantly reduced, while postprandial adiponectin, GLP-1, insulin, and serum insulin-like growth factor 1 (IGF-1) levels are significantly increased. Increased adiponectin levels are associated with changes in total fat mass (*R* = −0.64, *p* = 0.003) and reduced risk of atherosclerosis [44]. The increase in GLP-1 levels after weight loss surgery can reverse obesity-induced endothelial dysfunction, restore the endothelial protective properties of HDL [45], and reduce insulin resistance. The reduction in insulin resistance and IGF-1 consequently decreases the risk of common carotid intima-media thickness (ccIMT) in young, morbidly obese patients [46]. Serum Hsp60 is elevated in morbidly obese patients, but decreases after surgical weight loss. As a predictor of cardiovascular disease, Hsp60 is associated with inflammatory markers, i.e., ApoB/ApoA1 and cholesterol/HDL ratios. Hsp60’s association with cardiovascular risk as a proinflammatory adipofactor suggests that Hsp60 may be a molecular link between adipose tissue inflammation and the development of cardiovascular disease [47].

Laparoscopic sleeve gastrectomy can significantly reduce weight and control blood pressure in obese patients. Studies have shown that the remission rate of hypertension varies between 60% and 70% in the year after weight loss surgery and may even reach 90% in long-term follow-up [48,49]. Inna et al. found that 63% of patients reduced their antihypertensive medications and 23% discontinued treatment after weight loss in a 1 year follow-up [50].

### 4.5. Effects on NAFLD

The impact of bariatric surgery on the course of NAFLD in obese individuals has been extensively reported [51,52,53,54]. Studies indicate that NAFLD likely causes various cardiovascular and hepatic complications despite its seemingly benign nature. Bariatric surgery is a potential method of inhibiting disease progression and altering the expected natural history of NAFLD. In order to further understand how bariatric surgery may curb NAFLD progression, it is important to understand the biomarkers that drive NAFLD and its vast impacts. While the liver is mainly affected, a large portion of the gastrointestinal system is inhibited by the presence of various biomarkers that disrupt chemical and endocrine functionality. These biomarkers consist of, but are not limited to, cholesteryl ester transfer protein (CETP) [55], neurotensin (NT) [56], and vitamin D [57]. CETP accompanies NAFLD progression through metabolic liver inflammation. High levels of NT have been associated with high rates of NAFLD, cardiovascular disease, T2DM, and obesity. Insufficient or deficient vitamin D levels are indicative of NAFLD progression in association with fibrinogen levels, PCR levels, and T2DM [57].

## 5. Endocrine Function after Bariatric Surgery

Endocrine function changes after bariatric surgery. This may be attributed to the physically different pathways that nutrients take after surgery, which alters the gastrointestinal tract’s structure. This theory suggests that the pancreas may have heightened sensitivity and adaptability with the creation of new digestive pathways. The changes to endocrine function are vast with bariatric surgery, including changes in osteoporosis rates and hypoglycemia [58]. Many studies indicate that glucagon-like peptide-1 (GLP-1) is a primary biomarker present after bariatric surgery. Other research shows increased bone metabolism via heightened osteoclast activity [35] and hypothalamic inflammation [29] as a mechanism of increased weight loss. Further analysis shows that hormones levels of ghrelin, cholecystokinin (CCK), peptide YY (PYY) [59], thyroid-stimulating hormone (TSH), free triiodothyronine-3 (FT3), and triiodothyronine-3 (T3) fluctuate post surgery, with decreases in TSH, FT3, and T3 [60]. GLP-1 is noted as the hormone that connects endocrine function to glucose levels. Furthermore, obesity is largely affected by insulin sensitivity, beta-cell function (pancreatic origin), incretin response, gut microbiota, and fat and glucose metabolism post bariatric surgery [61]. However, a number of aforementioned mechanisms remain heavily debated.

### 5.1. GLP-1

GLP-1 is a peptide hormone and neurotransmitter that has numerous metabolic and nonmetabolic effects such as its ability to enhance B-cell function [62]. The rapid entry and absorption of nutrients in the distal small bowel induces GLP-1 increases (up to threefold), which is secreted by L cells from the gut. Increased GLP-1 levels improve beta-cell function and insulin sensitivity [63]. The postprandial levels of GLP-1 are enhanced in patients after SG or RYGB [64,65]. Although most research on GLP-1 has focused on glucose metabolism, the function of GLP-1 extends beyond glycemic control. GLP-1 has a documented dose-dependent effect on satiety. GLP-1 promotes satiety, potentiates insulin release, and suppresses glucagon release in response to nutrient ingestion [66]. During weight loss after bariatric surgery, GLP-1 may play important roles in changes to metabolism. While the mechanisms remain to be determined, one study using animal and human models showed a significant postprandial rise in GLP-1 and incretin, a peptide hormone. Furthermore, multiple studies correlated bariatric surgery and the rise in GLP-1 with metabolic improvements post surgery [67,68,69,70,71,72,73,74], although the pharmacological mechanism remains unclear [74]. GLP-1 and its analogs are popular T2DM medications because they decrease appetite, thereby decreasing food intake and increasing feelings of satiety. The physical alterations of the gut as a result of RYGB and other bariatric surgery procedures result in GLP-1 level changes, which are sensitized by an altered digestive tract and nutrient absorption process [70,72]. These findings point to a possible mechanism of relative glycemic control and weight loss found post surgery involving GLP-1, glucose homeostasis, and T2DM remission.

In patients with NASH, GLP-1 decreases de novo lipogenesis, lipolysis-induced FFA levels, and triglyceride-derived toxic metabolites [75]. It is undeniable that there is a close relationship between GLP-1 and lipid metabolism, metabolic improvements, and T2DM remission.

### 5.2. FGF19 and FGF21

Fibroblast growth factors (FGF) consist of at least 22 different groups with a variety of functions. Their functions include, but are not limited to, glucose and lipid metabolism [76,77,78,79,80], angiogenesis [81], epithelial and endothelial wound repair [81,82], and cell growth and differentiation [82]. They are also structurally capable of acting as hormones. According to the evidence, increases in FGF are inevitable post bariatric surgery and catalyze weight loss in obese and nonobese patients [83]. Interestingly, other evidence indicates that the effect of FGF and its amount in vivo is surgery-specific. Due to its relationship with increased macronutrient ingestion and fast glucose delivery rates to the liver, FGF levels may be able to serve as a potential biomarker that indicates weight loss after bariatric surgery [76,77,78,79,80,81,82].

In humans, the hormone-like members of the FGF family include FGF-15/19 and FGF-21 [84]. Metabolism of FGF-19 and FGF-21 involves regulation of bile-acid biosynthesis, glucose metabolism, and lipid metabolism in beta cells of islets, liver, and adipose tissue [85,86]. FGF-21 was also shown to have a protective effect against hepatic steatosis in a mouse model [87]. The levels of plasma FGF-19 and FGF-21 were analyzed in 28 patients who underwent RYGB (*n* = 16, two men and 14 women; age, 43.7 ± 1.8 years; body mass index, 45.5 ± 1.4 kg/m^2^) or LAGB (*n* = 12, one man and 11 women; age, 47.7 ± 3.8 years; body mass index, 46.6 ± 2.3 kg/m^2^) procedures [77]. They found that both RYGB and LAGB surgeries induced an increase in postprandial plasma FGF-19 concentrations. They also found that plasma FGF-21 surpassed baseline levels by about threefold after RYGB surgery. Similar results were found in research conducted by Crujeiras et al. [88]. Bariatric surgery induces FGF-15/19 expression in the distal small intestine through direct stimulation of FXR by bile acids. Additionally, FGF-15/19 acts in an endocrine-like manner to improve metabolic function on insulin target tissues. FGF-15/1 may also induce hepatic FGF-21 expression, which potentiates its beneficial effects on metabolic activity [89].

### 5.3. Pancreatic Peptide YY

PYY, also known as pancreatic peptide YY3–36, is a peptide gut hormone encoded by the PYY gene in humans. PYY is a short (36 amino acids) peptide released by L enteroendocrine cells in the distal small intestine and colon in response to feeding [90]. The main role of PYY involves the central regulation of appetite [91]. PYY mediated weight loss after bypass surgery in a mouse model [92] and increased PYY was observed in patients after bariatric surgery [93]. Interestingly, patients with NAFLD had significantly increased fasting PYY levels [94]. Overall, due to the close relationship between PYY and both weight loss after bariatric surgery and NAFLD, its role in metabolism is worthy of deeper exploration.

PYY is also thought to regulate glucose homeostasis [95]. Identification of the factors that increase PYY expression may benefit not only glucose monitoring but also possible novel targets for achieving glucose control [96]. Further research in this pathway may offer more nonsurgical treatments for T2DM. Past studies have shown a significant metabolic change after bariatric surgery [97]. After PYY was administered to patients post surgery, there were noticeable stabilizations of glucose, cholesterol, glucagon, and triglyceride levels. The conferred metabolic stabilization is relevant to weight stabilization. Although more research is necessary, it is plausible that the elevation of PYY levels is associated with stabilization of glucose levels, metabolism, and weight, which directly impact both the rates and the complications of obesity and diabetes. Some studies suggest that any differentially altered hormone levels, such as PYY, are the result of changing gastrointestinal anatomy post surgery [98]. However, whether PYY levels catalyze glycemic control, and whether its origin is endogenous or exogenous (as well as the relevance of exogenous vs. endogenous PYY levels) are inconclusive at this time.

## 6. Other Effects of Bariatric Surgery

### 6.1. Gut Microbiota

Numerous microorganisms exist on the surface of skin and mucosal linings in the human body. The human intestinal tract is colonized by a unique collection of microbes, termed the human gut microbiota. There is evidence that the specific composition of the intestinal microbiome is altered by NAFLD [99,100]. Additionally, studies showed that SG and RYGB led to decreases in Firmicutes and increase in Bacteroidetes and Proteobacteria after SG and RYGB [101,102,103,104,105,106]. These changes are likely to play a positive role in metabolic diseases [38,107,108,109]; hence, it cannot be dismissed that bariatric surgery improves NAFLD by affecting the gut microbiota.

### 6.2. Bile Acids

Bile acids (BAs) are synthesized from cholesterol, primarily in hepatocytes, and transported to the gallbladder for storage. Then, along with other biliary constituents, BAs are emptied into the small intestine where they function in the emulsification and absorption of dietary fat, cholesterol, and fat-soluble vitamins. After reaching the terminal ileum, BAs are almost completely (∼95%) absorbed by an active uptake mechanism [110]. BAs, as signaling molecules, can improve NAFLD through multiple mechanisms such as glucose and metabolism regulation and diminishing metabolic inflammation. [111]. Bariatric surgery can increase BA levels, change BA composition, decrease hepatic steatosis, and ameliorate insulin resistance [112]. This may be due to surgery’s modification of the pathway BAs take from the gallbladder to the ileum [113]. However, compared to the general population, the risk of symptomatic gallstone disease was increased fivefold after bariatric surgery techniques of VBG, AGB, and RYGB [114,115,116], which may have been caused by the changes to gallbladder bile composition [117]. Recent research showed that female gender and rapid weight loss are major risk factors [118,119].

## 7. Complications of Bariatric Surgery

### 7.1. Osteoporosis

Increased osteoclast activity has been demonstrated as a prevalent post-bariatric-surgery phenomenon that directly impedes bone homeostasis. Multiple studies indicate that this increase in osteoclast activity can lead to complications including osteoporosis, hypovitaminosis D, and recurrent bone fractures [120,121,122]. The mechanism via which this occurs has been examined, and several conclusions have been reached, such as high rates of calcium malabsorption as a result of the overall malabsorption of key nutrients post surgery. Various studies explored these mechanisms further by analyzing the physical elimination of nutrients and various malabsorption phenomena that are directly attributable to the surgeries performed.

### 7.2. Diarrhea

Diarrhea is a common complication after RYGB and BPD [123,124,125], which has a major impact on quality of life as well as nutrient and vitamin absorption [126,127]. There are several causes of diarrhea after bariatric surgery: short bowel syndrome, dumping syndrome, and inflammatory bowel disease, to name a few. These diseases are challenging to distinguish, making diagnosis difficult [128]. Early identification of the causes of diarrhea and targeted treatment would prove beneficial.

### 7.3. Iron Deficiency and Anemia

Different factors contribute to the development of iron deficiency after bariatric surgery including reduction of iron intake, hydrochloric acid secretion levels, and absorptive surface area [129,130]. The incidence of anemia following bariatric surgery has been reported to be as high as 74%, mostly ascribed to iron deficiency [131]. Obese patients are also more likely to suffer from iron deficiency and anemia, especially females [132,133]. Therefore, postoperative iron supplementation and follow-up are very important.

## 8. Conclusions

Consequences of obesity are vast, including NAFLD, insulin resistance, and metabolic disorders. Scientific studies suggest that bariatric surgery is an effective treatment capable of curbing rates of obesity, cardiometabolic rates, and various liver morbidities. Bariatric surgery can potentially halt or reverse progressions of NAFLD and T2DM, with evidence suggesting that it can also lead to T1DM remission. In addition, bariatric surgery affects cardiovascular disease and cerebrovascular disease. Many studies have promoted the benefits of bariatric surgery through measurable changes in eating habits and weight loss. Complications that can arise from bariatric surgery include osteoporosis, altered gut microbiota, and faulty bile-acid regulation. These complications are likely due to new physical and chemical manifestations of the operations. As the risks are not outweighed by the benefits, the current literature suggests bariatric surgery as an optimal treatment to limit disease progression of obesity and associated comorbidities.

## Figures and Tables

**Figure 1 biomolecules-11-01582-f001:**
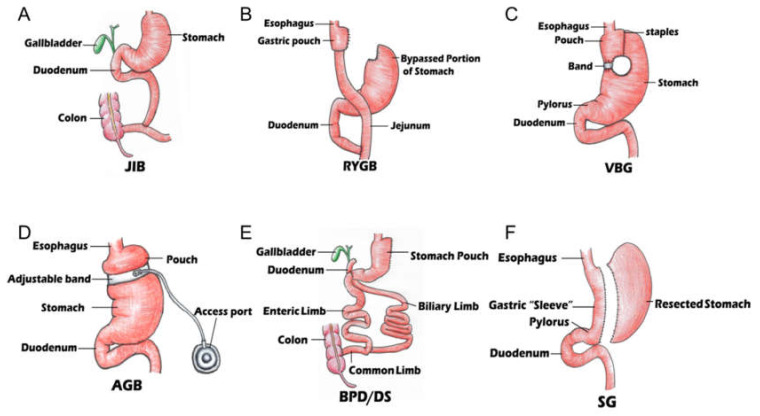
Different types of surgeries: (**A**) jejunoileal bypass surgery (JIB); (**B**) Roux-en-Y gastric bypass (RYGB); (**C**) vertical banded gastroplasty (VBG); (**D**) adjustable gastric band (AGB); (**E**): biliopancreatic diversion with duodenal switch (BPD/DS); (**F**) sleeve gastrectomy (SG).

**Table 1 biomolecules-11-01582-t001:** Different types of surgeries and effects.

	JIB	RYGB	VBG	AGB	BPD&DS	SG
Origin	1953	1977	1982	1986	1998	2005
Gastric Body	✓	✓	✓	✓	✗	✗
Bypass	✓	✓	✗	✗	✗	✗
Biliopancreatic diversion	✗	✗	✗	✗	✓	✗
Bowel anastomosis	✓	✓	✗	✗	✓	✗
Weight loss(36M)	-	90 Lbs/41 kg	71 Lbs/32 kg	-	117 Lbs/53 kg	-

## Data Availability

Not applicable.

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
