# Peer review of "Effect of Bariatric Surgery on Metabolic Diseases and Underlying Mechanisms"

_biomolecules, 2021, doi:10.3390/biom11111582_

Round 1

Reviewer 1 Report

Ji Y and co-authors reviewed an interesting issue regarding patients after bariatric surgery. They structured the manuscript by focussing first on non-alcoholic fatty liver disease -NAFLD- and later on some benefitial points afeter bariatric surgery such as eating habits, weight loss, diabetes, cardivascular and cerebrovascular diseases and modifications on NAFLD. In the third part of the review authors discuss the role of GLP-1, FGF family and PYY. Finally, some other effects of bariatric surgery and complications are mentioned.

However, I have some concerns regarding the organization of the manuscript. 

First, I would recommend a title more suitable to the points discussed in the manuscript. The current title opens expectatives that are not found in the text or only found as small paragraphs as other discussed points. 

Authors should better define the scope of this review. If the review aims to discuss clinical issues regarding bariatric surgery, I agree that this is the text. If not, as indicated in the title, the paragraphs should include more information about molecular pathways.

May the authors explain the reason to include types of bariatric surgeries? I am unable to find an explanation for it.

Author Response

Thank you for the valuable comments and suggestions. According to your recommendation, we have changed our title to: “Effect of Bariatric Surgery on Metabolic Diseases and Underlying Mechanisms”, which better reflects the focus of the current review.

In this review, we discussed how bariatric surgery effects non-alcoholic fatty liver disease, diabetes, cardiovascular disease, and cerebrovascular disease. We also describe the molecular mechanisms of the altered endocrine function involving GLP-1, Hsp60, leptin, Adiponectin, FGF19 /FGF21, and Pancreatic Peptide YY. We have clarified the scope of this review in both the abstract and introduction. 

The different types of bariatric surgery were included since the unique operations have varying effects on the body’s metabolic function in accordance with their final anatomic characteristics. Hence, the variations of benefit per surgery type and their mechanisms were introduced and discussed. We have clarified the relevance of describing the types of bariatric surgeries in the revision and have relocated it to a more coherent location (section 3.1).

Reviewer 2 Report

The study presents effects of bariatric surgeries. The manuscript should be re-organised. 
The title of this review article does not match the content. Presentation of "beneficial effects of bariatric surgeries on cardiovascular and cerebrovascular diseases and diabetes" consists only small part of the whole manuscript. Considering that the authors also describe the complications and side effects of the bariatric surgeries, the manuscript title should be changed into for example: "Effects of bariatric surgeries".
The authors should not concentrate on NAFLD in "Introduction". Could you replace descriptions of bariatric surgeries (including Figure 1) and locate them in "Introduction"?
Please provide "Conclusions" of your study.

Author Response

Thank you for the valuable comments and suggestions. As mentioned above, we have changed our title to: “Effect of Bariatric Surgery on Metabolic Diseases and Underlying Mechanisms”, which better reflects the focus of the current review.

Based on your suggestions, the revision relocated the description of the different types of bariatric surgeries (including Figure 1) to an earlier portion of the paper and elaborated on its relevance (section 3.1). We have also appended a Conclusions section into the revision. 

Round 2

Reviewer 1 Report

The manuscript improved considerbly. I accept it with the present form.

Reviewer 2 Report

Dear Authors,

Thank you for attempting to address my concerns.